# Video Scratch Removal Method Based on Guided Diffusion Generation With Locally Constrained Generation

## Abstract

This paper proposes a video scratch removal method based on guided diffusion with locally constrained generation. By estimating optical flow between adjacent frames and integrating visual features, we construct a guidance map that encodes both temporal and spatial information. A relaxed mask is computed using a mean filter, which helps mitigate motion estimation errors while effectively distinguishing potential scratch regions.To further reduce motion-induced blurring during the restoration process, the proposed method leverages the guidance map—fused from temporal and spatial cues of neighboring frames—as a conditioning input to the diffusion model for restoring the current frame. In addition, the relaxed mask is used to constrain the generation to local regions, allowing uncorrupted areas to retain their original content.Experimental results demonstrate that our approach not only significantly improves restoration quality but also effectively reduces the likelihood of missing scratches.

## 1 Introduction

A vast number of old films and video archives with significant artistic and historical value still exist today. However, due to long-term environmental degradation and improper storage conditions, these invaluable visual records often suffer from irreversible image deterioration, including scratches, blotches, and tears. Such degradations severely compromise both the visual quality and the semantic integrity of the content. To preserve the cultural memory and historical value embodied in these legacy films, film restoration technologies have been widely adopted in archives, museums, and professional restoration institutions. Traditional restoration workflows, heavily rely on manual labor, which is time-consuming, labor-intensive, and inherently difficult to scale to large collections. This highlights the pressing need for developing automated, intelligent, and efficient restoration methods.

Among the various forms of degradation in deteriorated vintage films, scratches are among the most common and visually disruptive artifacts. This paper focuses on addressing the problem of scratch removal in old film footage. Despite recent advances in restoration techniques, several challenges remain. Due to the age and condition of analog film, their digitized counterparts often suffer from inherently poor image quality, characterized by low resolution, high levels of noise, and limited dynamic range Liu et al. (2024). Moreover, in scenes with rich textures or complex motion patterns, scratches are frequently entangled with the intrinsic texture structures within frames. This entanglement leads to indistinct scratch boundaries and uneven intensity profiles, making it difficult for traditional detection methods—typically based on simple brightness cues, geometric shapes, or local statistical features to accurately distinguish genuine scratch artifacts from fine image details. Recent methods Wan et al. (2022); Liang et al. (2022); Lin & Simo-Serra (2024) still exhibit significant limitations in addressing the scratch–texture confusion problem in such low-quality sources, often resulting in erroneous processing of complex background regions, as illustrated in Fig. 1.

Moreover, as a form of temporal information, old film restoration faces a critical challenge induced by inter-frame motion: visual consistency.Scratches in old films typically appear randomly, while camera motion and dynamic scenes cause continuous changes in the visibility of scratches and their occlusion relationships with the background throughout the video sequence.If the restoration algorithm lacks effective modeling of spatiotemporal context—processing frames in isolation or relying on unreliable motion estimation—the restored regions are highly prone to severe artifacts across consecutive frames, such as abrupt changes in color, brightness, texture, or boundary

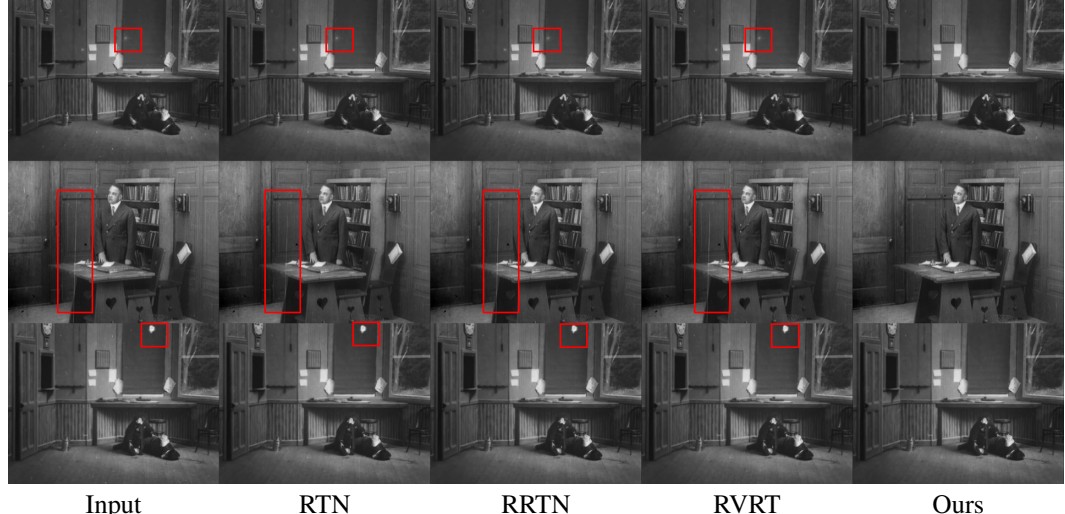

| Input | RTN | RRTN | RVRT | Ours |

Figure 1: In complex real-world scenes, textures and scratches often exhibit similar visual characteristics, causing models to easily confuse them and resulting in unsuccessful restoration of scratched regions. The above figure shows the restoration results of different methods trained on the same dataset, where our method demonstrates significantly better performance in addressing the confusion between textures and scratches.

misalignments.These dynamic artifacts are visually jarring and can severely undermine the temporal coherence of the video, sometimes causing more perceptual disturbance than the original static scratches.Among existing methods, traditional motion compensation techniques are constrained by the inaccuracies of optical flow estimation on the low-quality images typical of old films. Meanwhile, many approaches lack explicit guidance of visual features in the temporal dimension, leading to suboptimal performance in handling scenes with motion.

To address the dual core challenges of "texture-scratch confusion" and "visual consistency" in the restoration of low-quality old film videos, we propose a video scratch restoration method based on guided diffusion with local generation. Our key contributions are summarized as follows:

- We propose a spatiotemporal collaborative displacement-relaxed restoration strategy, which integrates inter-frame motion information and visual features to construct a motion-guided guidance map, enabling accurate modeling of spatiotemporal variations. The guidance map further provides scratch region priors to alleviate the problems of missing or incorrect restorations.

- We introduce a guidance-map-driven video diffusion restoration framework. Specifically, we design a temporally-conditioned diffusion model based on the guidance map to achieve fine-grained restoration of degraded frames, effectively mitigating motion-induced blur and artifacts.During training, the model learns constrained generation of scratch regions in high-dimensional feature space. During inference, a local generation strategy is employed to fully leverage temporal dependencies and visual cues across frames.

- We conduct experiments on both synthetic and real-world video scratch datasets. Results demonstrate that our method outperforms existing state-of-the-art approaches in both subjective visual quality and objective evaluation metrics, achieving superior performance and validating its effectiveness and practicality.

## 2 RELATED WORK

### 2.1 VIDEO RESTORATION

The task of scratch removal in old films can be regarded as a subcategory of video restoration. In this broader context, video denoising Monakhova et al. (2022); Zheng et al. (2023); Liang et al. (2022)

aims to remove noise from video frames. In contrast, scratches are structured artifacts, typically manifesting as linear or patch-like occlusions, and their restoration relies more heavily on semantic understanding and localized reconstruction.; video deblurring Kim et al. (2024); Shang et al. (2021); Pan et al. (2023) primarily addresses frame-wide blur caused by motion or defocus, whereas scratch removal involves localized damage that requires content-aware reconstruction methods; and weather degradation removal, such as desnowing Ren et al. (2017); Chen et al. (2023); Wu et al. (2024), dehazing Xu et al. (2023); Liu et al. (2022a); Van Nguyen et al. (2022), and deraining Yang et al. (2019); Ren et al. (2017); Zhang et al. (2023), primarily addresses global degradations caused by natural weather conditions, which typically follow well-defined physical models. Video inpainting Xu et al. (2019); Quan et al. (2024) is conceptually similar to scratch removal, aiming to fill occluded or missing regions in a temporally coherent manner. However, video inpainting typically deals with relatively large, regular-shaped holes within predefined regions. In contrast, scratches in old films are often slender, irregularly shaped, and intricately distributed, posing significant challenges for both detection and restoration.

## 2.2 OLD FILM RESTORATION

DeepRemaster Iizuka & Simo-Serra (2019) was the first framework to apply temporal convolutional neural networks for old film restoration. To create training data, the authors synthesized degradation by combining real noisy film clips with high-quality videos, and then applied further algorithmic corruption to simulate realistic film damage. This enabled the model to learn an effective representation of old film degradation. The method addresses old flim mixed degradations, including flicker suppression, colorization, and overall visual enhancement.

Subsequent work Wan et al. (2022) employed a recurrent transformer network to more effectively model temporal dependencies. The method leverages optical flow between frames to enforce temporal consistency and uses flow cues to identify potential scratch regions. These methods often incorporate perceptual loss Johnson et al. (2016) to boost visual quality. Lin & Simo-Serra (2024) further improved upon this approach by integrating the Swin Transformer Liu et al. (2022b) to strengthen global context modeling. In addition, it introduced a more effective scratch discrimination mechanism and adopted recursive structures to better handle challenging types of film degradation. Mao et al. (2025) selected Mamba as the backbone network and proposed MambaOFR. Their approach introduces degradation-aware prompting to dynamically adapt to complex and mixed degradations in old films. Additionally, a flow-guided mask deformable alignment module is designed to reduce the temporal propagation of structured artifacts.

Cai et al. (2023) addresses scratch restoration in printed photos by leveraging collaborative scratch and background context. The core component, the Scratch Contextual Assisted Module, adaptively learns pixel-wise correspondence within the mask by computing distances between masked-out and encoder features, enhancing restoration quality. Liu et al. (2024) proposes a Gaussian probability-based model with local adaptive one-dimensional Gaussian weighted segmentation and invalid region filtering to enhance scratch integrity and reduce false positives. By incorporating trajectory information, it reduces missed detections and replaces unstable Kalman filtering with a U-Net-based method for more robust trajectory prediction.

## 2.3 DIFFUSION MODELS

Recently, Diffusion models Xia et al. (2023); Zhu et al. (2023); Cao et al. (2025) have gained significant attention in restoration tasks, particularly in the field of image inpainting. Their ability to generate high-fidelity and realistic content through a gradual denoising process has made them highly effective for image restoration. These models excel at reconstructing complex textures and semantically meaningful structures from sparse or degraded inputs.

However, the application of diffusion models to video restoration remains in its early stages. In old film restoration, there is significant room for improvement. We aim to leverage the high-quality generation capabilities of diffusion models to specifically restore scratches in old films, achieving more realistic results.

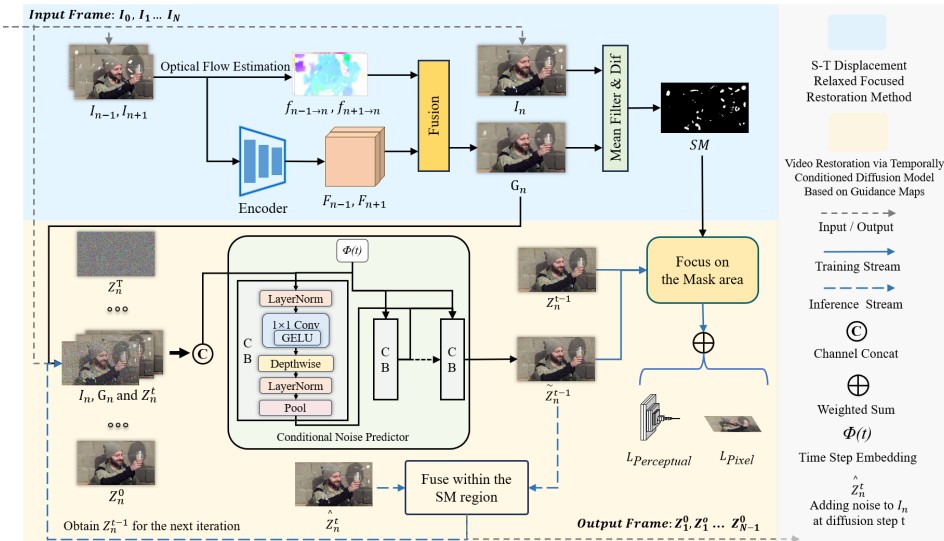

Figure 2: Overview of the proposed framework.

## 3 METHOD

The model takes as input a degraded video sequence containing $N + 1$ frames, denoted as $\{I_0, I_1, \ldots, I_N\}$. For each frame $I_n$, $n \leq N$, its adjacent neighboring frames $I_{n-1}$ and $I_{n+1}$ are used to compute a spatiotemporally coordinated guidance map $G_n$ and the corresponding scratch region mask $M_n$, which indicate the damaged areas that require restoration in the current frame. Based on this conditional information, a temporal conditional diffusion model performs noise prediction and reconstruction for the scratch regions on a frame-by-frame basis. The model outputs a set of $N$ repaired frames $\{Z_1^0, Z_2^0, \ldots, Z_N - 1^0\}$ corresponding to the input sequence, $Z_n^0$ denotes the repaired result of the input frame $I_n$. And each frame $Z_n^0$ is the restored clear image guided by $G_n$ and $M_n$ through the diffusion process. Our method has certain limitations, notably the first and last frames cannot be effectively restored. Pipeline overview in Fig. 2.

### 3.1 S-T DISPLACEMENT RELAXED FOCUSED RESTORATION METHOD

In most degraded old movie videos, there exist many small scratches that are easily mixed with the texture information in the video content. To accurately identify the scratched regions in video frames, this paper proposes a guided mask generation strategy that fuses spatial visual features with temporal motion information. This method utilizes the visual features and bidirectional optical flow information of the neighboring frames $I_{n-1}$ and $I_{n+1}$ around the current frame $I_n$ to construct a guidance map $G_n$ for $I_n$, and generates a scratch region mask by calculating the difference between this guidance map and the current frame. However, motion estimation between adjacent frames is often accompanied by some estimation errors, which affect the mask generation. Therefore, a mean filtering operation is introduced before the difference calculation to reduce the impact of motion estimation errors and generate a more reliable mask.

To achieve this goal, we introduce an optical flow estimation model to capture the motion information between frames, which is then used to predict the intermediate frame:

$$f_{n-1 \to n}, f_{n+1 \to n} = \text{Flow}(I_{n-1}, I_{n+1}), \tag{1}$$

where $I_{n-1}$ and $I_{n+1}$ denote the adjacent frames before and after the current frame $I_n$, respectively. The function *Flow* estimates the optical flow between frames. $f_{n-1 \to n}$ and $f_{n+1 \to n}$ represent the optical flow from frame $n - 1$ and frame $n + 1$ to the current frame $n$.

Thanks to some recent advance Wan et al. (2022), it has been shown that using only optical flow is still insufficient to accurately localize potential scratch regions. Therefore, we aim to incorporate

both optical flow and visual features to guide the detection of scratches. Specifically, we further compute the visual features from adjacent frames and fuse them with the optical flow to generate a guidance map:

$$G_n = \mathcal{G}(F_{n-1}, F_{n+1}, f_{n-1 \to n}, f_{n+1 \to n}), \tag{2}$$

where $F_{n-1}$ and $F_{n+1}$ denote the visual features of the adjacent frames, and $\mathcal{G}$ is the function for fusing optical flow and visual features. $G_n$ represents the guidance map for the $n$-th frame, which encodes both temporal and spatial information across neighboring frames. This guidance map is subsequently used as a conditioning input for generating restored frames, with related details provided in the following sections.

Since the guidance map may contain deviations introduced by motion estimation, we apply a mean filter to smooth the inter-frame motion. The filtered frames are then used to calculate the pixel-wise difference $D_n$ between the input and the guidance map:

$$D_n = |\text{MeanFilter}(I_n) - \text{MeanFilter}(G_n)|, \tag{3}$$

The resulting pixel difference is used to generate a binary mask of potential scratch regions:

$$M_n(x, y) = \begin{cases} 1, & D_n(x, y) > \tau, \\ 0, & \text{otherwise.} \end{cases} \tag{4}$$

$M_n(x, y)$ is a binary mask indicating whether pixel $(x, y)$ in frame $n$ is marked as a scratch (1) or a normal region (0). $\tau$ is a predefined threshold that distinguishes scratch regions from normal areas. The resulting spatiotemporal mask is used to guide the subsequent frame restoration process.

## 3.2 VIDEO RESTORATION VIA TEMPORALLY CONDITIONED DIFFUSION MODEL BASED ON GUIDANCE MAPS

Diffusion models exhibit strong generative capabilities and have been widely adopted in the image restoration domain. However, their capacity for temporal modeling remains limited. To address this issue, we build upon a spatiotemporal co-guidance mechanism for mask generation and further incorporate a guidance map $G_n$ to enhance the diffusion model's ability to capture temporal dependencies. Simultaneously, the mask serves to guide the identification of potential scratch regions. For high-fidelity reconstruction of the corrupted areas, we employ ConvNeXt as the noise prediction network. This network adopts a residual architecture, combined with deep convolutional layers, layer normalization, and GELU activation functions, thereby providing enhanced spatial modeling capabilities.

At each reverse diffusion step $t$, the model takes the current diffusion state $Z_n^t$ as input and integrates the guidance map $G_n$ and the original image $I_n$ as conditional inputs to predict the noise components within the masked region. During training, $Z_n^0$ is treated as the original clean image, while $Z_n^t$ is progressively generated from $Z_n^0$ through a forward diffusion process defined as:

$$Z_n^t = \sqrt{\alpha_t} Z_n^0 + \sqrt{1 - \alpha_t}\epsilon, \quad \epsilon \sim \mathcal{N}(0, \mathbf{I}) \tag{5}$$

During training, $Z_n^0$ denotes the undistorted image of the $n$-th frame, serving as the starting point of the diffusion process. $Z_n^t$ represents the corrupted image at timestep $t$, obtained by injecting noise into $Z_n^0$ over $t$ steps. The noise term $\epsilon$ follows a standard multivariate Gaussian distribution, i.e., $\mathcal{N}(0, \mathbf{I})$, simulating random perturbations in the latent space.

To improve the model's sensitivity to scratch regions during training, we incorporate the mask information $M_n$ into the loss function, thereby encouraging the network to focus more explicitly on the reconstruction quality within the damaged areas. Specifically, we impose explicit constraints in both pixel space and perceptual feature space on the masked regions, guiding the model to prioritize the semantic recovery of structurally incomplete areas. This design ensures that local fidelity in scratch

regions is optimized without compromising global consistency. The detailed formulation of the loss function will be presented in a subsequent section.

During inference, to ensure the stability of the generated results and preserve the content outside the masked regions, we propose a mask-based local update strategy. At each diffusion step, the model updates only the masked region $M_n$ by predicting its noise component, while maintaining the unmasked region $(1 - M_n)$ in its original noisy state:

$$Z_n^t = (1 - M_n) \odot \hat{Z}_n^t + M_n \odot \tilde{Z}_n^t,$$ (6)

where, $Z_n^t$ denotes the fused estimation of the $n$-th frame at diffusion step $t$. $Z_n^t$ is the result of the forward diffusion applied to the degraded frame, while $\tilde{Z}_n^t$ is the predicted output from the noise estimation network at the previous step. $M_n$ is the corresponding binary mask indicating the scratch regions. The final output at each step is represented by $\hat{Z}_n^t$, where, at $t = 0$, the non-scratch regions retain their original input values, and the scratch regions are reconstructed by the model. This strategy achieves a weighted fusion of restored and original content via the mask, ensuring a balance between local restoration and global visual consistency.

### 3.3 LOSS FUNCTION DESIGN

To effectively restore scratched regions in video frames and enhance the overall quality of the inpainting results, we incorporate two complementary loss functions during training: a perceptual loss and a pixel-wise L1 loss. The perceptual loss encourages semantic consistency in the reconstructed images, while the L1 loss ensures the accuracy of low-level details. The combination of these losses enables the model to preserve structural and textural fidelity while increasing similarity to the ground-truth images.

The perceptual loss evaluates the semantic similarity between the reconstructed image and the ground-truth image by comparing their representations in a high-level feature space. Specifically, we leverage a pre-trained VGG network to extract intermediate feature maps and compute the L2 distance between corresponding layers. The perceptual loss is formally defined as:

$$\mathcal{L}_{\text{perceptual}} = \sum_{l \in L} \frac{1}{N_l} \left\| \phi_l(\hat{I}_t) - \phi_l(I_t) \right\|_2^2$$ (7)

where $\phi_l(\cdot)$ denotes the feature extraction function at layer $l$ of the VGG network, $L$ is the set of selected layers for perceptual loss computation, and $N_l$ is the number of elements in the feature map at layer $l$. This loss function captures high-level semantic structures and textures, significantly improving the perceptual quality of the reconstructed images.

The pixel-wise L1 loss measures the absolute difference between the restored image and the ground-truth image at the pixel level, ensuring faithful reconstruction of fine-grained details. Compared to the L2 loss, the L1 loss imposes more balanced penalties across pixels, avoiding over-penalization of large errors. This makes it more robust in scenarios involving sparse noise or localized artifacts. The pixel-wise loss is defined as:

$$\mathcal{L}_{\text{pixel}} = \frac{1}{N} \sum_{i=1}^{N} \left\| \hat{I}_t(i) - I_t(i) \right\|_2^2$$ (8)

where $N$ denotes the total number of pixels in the image, and $\hat{I}_t(i)$ and $I_t(i)$ represent the values at the $i$-th pixel of the restored and ground-truth images, respectively. This loss promotes accurate reconstruction of local details and prevents over-smoothing or structural degradation.

To jointly optimize for both perceptual quality and pixel-level accuracy, the total training objective is formulated as a weighted sum of the two loss terms:

$$\mathcal{L}_{\text{total}} = \mathcal{L}_{\text{pixel}} + \lambda \mathcal{L}_{\text{perceptual}}$$ (9)

where, $\lambda$ is a hyperparameter that balances the contributions of the perceptual and pixel losses. By minimizing this total loss, the model is encouraged to restore visually plausible and semantically coherent content while preserving local textures and structures, thereby producing more natural and realistic inpainting results.

# 4 EXPERIMENTS AND RESULTS

To compare our method with state-of-the-art approaches, we trained all models, including ours, on the same dataset and conducted both quantitative and qualitative evaluations against three leading methods: RTN Wan et al. (2022), RRTN Lin & Simo-Serra (2024) and RVRT Liang et al. (2022). Additionally, we tested all trained models on a real-world public old film dataset.

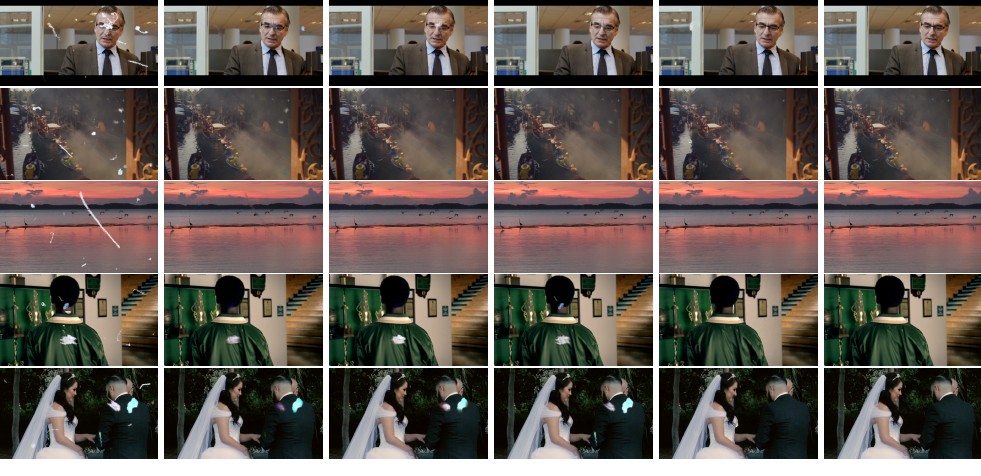

| Input | RTN | RRTN | RVRT | Ours | GT |

Figure 3: Compared to other methods, our approach consistently achieves superior restoration performance even in complex scenes.

At present, publicly available datasets specifically designed for video scratch removal remain scarce. Existing methods often rely on noise artifacts extracted from old films; however, such data typically contain various unrelated interference factors, limiting their effectiveness in the context of targeted scratch restoration tasks. To address this limitation, we collaborated with experts in film restoration to construct a dedicated dataset.

Approximately 1,000 scratch patches were manually extracted from real archival film material by professional restoration artists, ensuring both the authenticity and diversity of the scratch patterns. These small scratch images were then subjected to random transformations, including deformation, rotation, and composition, to generate 1,000 large-scale images containing varying quantities and types of scratches. Using custom Photoshop scripts, these synthetic scratch maps were seamlessly blended into real video frames to simulate realistic scratch effects. For the underlying video data, we employed the Vimeo-90K dataset as the primary source.

In this work, we utilize the triplet interpolation subset of the Vimeo-90K dataset. The Vimeo-90K dataset Xue et al. (2019) is a large-scale video collection consisting of approximately 90,000 high-quality video clips, each comprising hundreds of frames. It is widely used in various video processing tasks such as video super-resolution and denoising. The triplet interpolation subset provides intermediate frames between adjacent video frames. Given two input frames, $I_{n-1}$ and $I_{n+1}$, the dataset also provides the interpolated intermediate frame $I_n$. Our model architecture requires three input frames: the preceding frame $I_{n-1}$, the current frame $I_n$, and the succeeding frame $I_{n+1}$. Thus, this triplet subset is particularly suitable for training and evaluating video scratch inpainting models while remaining compatible with general video restoration frameworks.

It is important to note that separate batches of synthetic scratch images were used for training and testing. Specifically, the large-scale scratch datasets were partitioned to ensure a clear separation

between the training and testing sets. This design ensures that the model is evaluated on unseen data, thereby promoting reliable generalization and robust performance assessment.

## 4.1 QUANTITATIVE EVALUATION

To quantitatively evaluate our method, we conduct experiments on a synthetic dataset using widely adopted image quality metrics, including PSNR, SSIM, LPIPS and FID. The results are presented in Table. 1. As shown, our method achieves the best performance across all four metrics.

| Method | PSNR↑ | SSIM↑ | LPIPS↓ | FID↓ |
|--------|-------|-------|--------|------|
| RTN | 34.7612 | 0.9830 | 0.0333 | 22.2231 |
| RRTN | 34.4915 | 0.9829 | 0.0325 | 21.8555 |
| RVTN | 34.6248 | 0.9806 | 0.0317 | 22.3697 |
| Ours | **37.3129** | **0.9876** | **0.0160** | **16.4844** |

Table 1: Quantitative comparison on the synthetic dataset across four metrics: PSNR, SSIM, LPIPS, and FID. Our method achieves the best performance across all metrics.

| Method | L1↓ | IoU↑ | F1↑ | Acc↑ |
|--------|-----|------|-----|------|
| RTN | 0.2399 | 0.1148 | 0.1869 | 0.6956 |
| RRTN | 0.0221 | 0.4620 | 0.6110 | 0.8612 |
| Ours | **0.0196** | **0.4913** | **0.6408** | **0.8971** |

Table 2: Comparison of different methods for scratch region identification. Our method achieves the best performance across all metrics.

To verify the effectiveness of our proposed temporally-relaxed mask, we compare it with other approaches designed for identifying potential scratch regions. Taking the ground-truth scratch regions as reference, we evaluate the performance using four metrics: L1 score, IoU, F1 score, and pixel-level Accuracy. The results are summarized in Table. 2. Our method outperforms the others across all metrics, demonstrating its superior ability in scratch region localization and segmentation.

## 4.2 QUALITATIVE EVALUATION

For the quantitative evaluation, we conducted comparisons on both the synthesized data and real old film. In Fig. 1, we selected four degradation cases that are difficult to detect, and the results show that our method effectively reduces missed restorations. In Fig. 3, we used the synthetic dataset and selected five complex scenes with significant motion interference. The experimental results demonstrate that our approach significantly improves visual consistency. To demonstrate the effectiveness of our method in distinguishing scratches, we selected representative samples from the synthetic dataset. As shown in Fig. 4, our method can more accurately identify potential scratch regions.Since RVRT does not provide explicit guidance for scratch regions, we only compared our method with the other two approaches.

## 5 ABLATION STUDY

Table. 3 presents the results of our ablation study, highlighting the contributions of key components in our method.To better assess restoration quality in degraded regions, we further calculate PSNR and SSIM exclusively within the GT mask area (denoted as $PSNR_M$ and $SSIM_M$). The base setting directly inputs consecutive frames into the diffusion model without any guidance, resulting in suboptimal performance due to the lack of effective motion modeling. The w/o L, MG setting removes both the mask-guided loss constraint and the local generation mechanism, leading to poorer restoration in damaged areas. Introducing either the loss constraint (w/o MG) or the mask guidance (w/o L) yields noticeable improvements, especially in scratch-specific metrics.

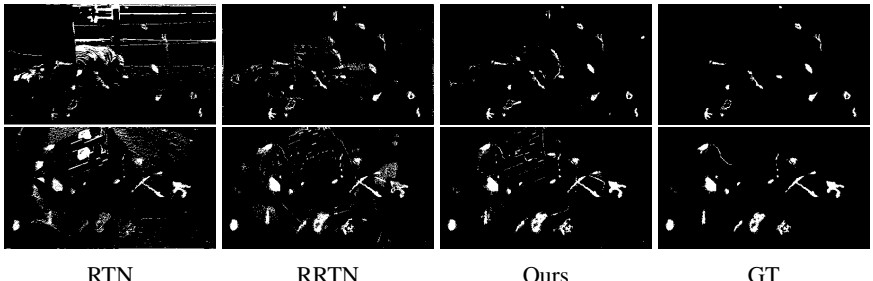

| RTN | RRTN | Ours | GT |

Figure 4: The comparison of mask generation results across different methods demonstrates that our approach can more accurately identify the scratch regions.

| Method | PSNR↑ | PSNR$_M$↑ | SSIM↑ | SSIM$_M$↑ |
|--------|-------|-----------|-------|-----------|
| base | 21.4951 | 22.2896 | 0.6882 | 0.8647 |
| w/o L, MG | 37.6563 | 38.2032 | 0.9793 | 0.9883 |
| w/o L | 37.4488 | 39.0587 | 0.9793 | 0.9912 |
| Ours | **38.1208** | **39.1311** | **0.9801** | **0.9912** |

Table 3: Ablation studies on the synthetic dataset.

## 6 CONCLUSION

This paper proposes a novel video scratch restoration method: Video Scratch Removal Method Based on Guided Diffusion Generation. The method leverages optical flow and visual features between adjacent frames to generate a guidance map that encodes both temporal and spatial information. A relaxed mask is then computed using a mean filter, effectively distinguishing potential scratch regions. By conditioning the diffusion process on this guidance map, the model restores degraded frames with higher accuracy and visual consistency, outperforming existing approaches.

Despite the notable improvements in scratch restoration, our method still has some limitations: The diffusion model incurs high computational overhead; It is currently unable to restore the first and last frames in a sequence. This could be a direction for future optimization.

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

## A  APPENDIX

You may include other additional sections here.

