# OpenReview forum: "Video Scratch Removal Method Based on Guided Diffusion Generation With Locally Constrained Generation"
_ICLR.cc/2026/Conference — Submitted to ICLR 2026_

### Official Review · Reviewer_UUV5 · 2025-10-30

**Soundness:** 2
**Presentation:** 2
**Contribution:** 2
**Rating:** 4
**Confidence:** 4

**Summary:**

This paper tackles fixing scratches in old films using a guided diffusion model. The core idea is to *constrain* the generation just to the bad spots.

To do this, you build a "guidance map" by mixing optical flow (for motion) and visual features from nearby frames. Then, you create a "relaxed" mask (using a mean filter) to find the *potential* scratch areas. The diffusion model is then fed this guidance and the mask, making it focus its fixing magic *only* on the damaged parts while leaving the clean parts alone.

Your experiments on both synthetic and real scratch datasets show this approach beats the current state-of-the-art, and you've got ablation studies to back it up.

**Strengths:**

* **Great Ablation Study**: Table 3 is really helpful. You did a thorough job showing exactly how much the "mask-guided loss" and "local generation" parts each contribute to the final performance. It's nice to see what's *actually* driving the improvements.
* **Strong Results**: The numbers look good. You're beating SOTA across the board on all those metrics (PSNR, SSIM, LPIPS, FID, etc.) in Tables 1 and 2. The visual examples (Figs. 1, 3, 4) also really sell it—they look much better than the baselines, especially on tricky parts.

**Weaknesses:**

1.  **You Don't Really Show Where It Fails**: You mention in the conclusion that it messes up on the first/last frames and is computationally heavy, but you don't actually *show* us these limitations. I really wanted to see what happens with very challenging motion, or how bad the output gets when the optical flow calculation is just plain wrong.
2.  **What About Temporal Consistency (i.e., Flicker)?**:
    * You use optical flow to get temporal cues, but then you never report any metrics that *actually measure* temporal coherence (like warping error or temporal SSIM).
    * You only restore the *middle* frame in each triplet. I'm worried this could lead to errors propagating from one frame to the next, or just cause flickering at the edges of the repaired spots. You don't really discuss this risk.
3.  **Does It Work on *Real* Old Movies?**: You built a nice synthetic dataset, and you show a few real-world examples, but there's almost no quantitative data or even a good description of the real videos. How do we know this generalizes? What happens when it sees real film grain, noise profiles, or types of motion that weren't in your training data?

**Questions:**

1.  Can you add more ablation studies for the $\lambda$ (lambda) hyperparameter in your loss function?
2.  Did you *try* to measure temporal consistency (like with warping error or temporal SSIM)? If not, why did you decide against it?
3.  For the qualitative part, did you run any human studies (i.e., ask people which video looked better)? Or, failing that, can you just provide more real-world examples, especially failure cases or borderline ones where it *almost* worked?

---

### Official Review · Reviewer_33AZ · 2025-10-30

**Soundness:** 2
**Presentation:** 2
**Contribution:** 2
**Rating:** 2
**Confidence:** 3

**Summary:**

This paper addresses two major challenges in scratch removal for old film videos: the difficulty of distinguishing scratches from textures leading to false detections, and the temporal inconsistency across restored frames. To tackle these issues, the authors propose a video scratch removal method based on a guided diffusion framework with spatio-temporal collaboration. The approach first constructs a guidance map by fusing optical flow and visual features from neighboring frames, and generates a robust restoration mask using mean filtering and pixel-wise differences to accurately locate scratch regions. Then, a mask-guided diffusion model is employed, which during inference updates only the masked areas while preserving uncorrupted regions, thereby ensuring high restoration quality and effective temporal coherence.

**Strengths:**

The method shows strong performance on benchmark datasets, achieving high scores in metrics like PSNR, SSIM, and F1. It outperforms existing methods in both scratch detection and restoration quality.

**Weaknesses:**

1. The method proposed in the paper requires relatively intact first and last frames to work, which to some extent limits its applicability in real-world scenarios.
2. The proposed method relies on accurate optical flow estimation, and the mask generation may not be robust under large motions, which could directly affect the result of video reconstruction.
3. Identical scratches appearing in both the previous and current frames may not be detected by optical flow estimation, as they would be perceived as consistent motion or structure, potentially leading to missed masks.
4. The paper contains numerous typos, lacks the title, and misses citations for some of the methods described.

**Questions:**

1. In the proposed method, accurately determining the spatial locations that need restoration is crucial. What implementation is used for the visual encoder, and what kind of information does it provide?
2. The diffusion model in the paper employs a noise addition and denoising process, supervised by pixel loss and perceptual loss. Is paired data required for training this diffusion model? Additionally, in the ablation study, why does removing the loss term not lead to a significant performance drop?

---

### Official Review · Reviewer_6sbs · 2025-10-31

**Soundness:** 3
**Presentation:** 3
**Contribution:** 3
**Rating:** 6
**Confidence:** 4

**Summary:**

The paper addresses video scratch removal for old films using a guided diffusion framework. It builds a spatiotemporal guidance map from optical flow and neighboring frame features and derives a relaxed mask to constrain local generation. A temporally conditioned diffusion model restores only masked regions while preserving clean areas. Experiments on a synthesized dataset and real old film clips show improvements over RTN, RRTN, and RVRT in both restoration quality and scratch localization.

**Strengths:**

- Clear problem focus with a coherent diffusion-based pipeline that fuses motion and appearance cues
- Local masked generation is well motivated and reduces texture scratch confusion
- Strong quantitative results against classical video restoration baselines with informative ablations

**Weaknesses:**

- Missing comparisons to recent diffusion-based video restoration models such as Upscale A Video CVPR 2024 and SeedVR CVPR 2025, which are important contemporary references
- No efficiency analysis, runtime peak, memory, and parameter counts are not reported, and the scalability to higher resolutions is unclear
• Temporal consistency evaluation is limited, reporting only per-frame metrics; should include perceptual temporal metrics such as **DOVER** and **tLPIPS** for a more comprehensive assessment

**Questions:**

1. Can the authors include results on diffusion-based video restoration methods (e.g., Upscale-A-Video, SeedVR) for direct comparison
2. Please report runtime, peak memory, and parameter statistics under identical hardware settings
3. Could additional temporal consistency metrics, such as DOVER and tLPIPS, be added to evaluate perceptual coherence

---

### Official Review · Reviewer_GaaC · 2025-11-01

**Soundness:** 3
**Presentation:** 3
**Contribution:** 2
**Rating:** 4
**Confidence:** 3

**Summary:**

I think the paper tackles old-film video scratch removal with a diffusion-based inpainting model guided by a spatio-temporal “guidance map” and a relaxed (mean-filtered) mask. Optical flow and neighbor-frame features are fused to form the guidance map; the diffusion model (ConvNeXt denoiser) is conditioned on the guidance plus the mask and only locally updates masked regions during reverse steps. Experiments use synthetic scratches blended into Vimeo-90K triplets and some qualitative real-film demos; the paper reports strong gains over RTN/RRTN/RVRT on PSNR/SSIM/LPIPS/FID and shows a small ablation.

**Strengths:**

(1)  I think the problem and the pipeline settings are very clear. The task (scratch removal) is well-motivated, and the pipeline—flow/feature-based guidance → mask → mask-local diffusion updates—is straightforward and sensible.

(2) Local generation idea is practical for avoiding collateral damage in clean regions; the eq. (6) “masked update” makes that explicit.

(3) Synthetic dataset construction is at least described (1k real patches → augment → composite onto Vimeo-90K triplets), which is valuable for reproducibility of the data side.

**Weaknesses:**

(1) The method is underspecified where it matters. The critical parts—what optical flow model, how you fuse flow + features (the function G), guidance-map architecture, mask threshold τ, mean-filter kernel/stride—are hand-wavy. The paper literally leaves G(·) as a symbolic box (eq. 2), and the flow choice is never named. This makes reproducing and assessing why the method works hard.

(2) Evaluation is small and slanted toward easy setups. Nearly all quantitative results are on synthetic data (scratches pasted on Vimeo-90K) with classic 2D metrics (PSNR/SSIM/LPIPS/FID). That’s fine for a start, but where are temporal metrics (t-LPIPS, warping error, FVD) and real-film quant? You claim temporal coherence gains in text, but never measure them.

(3) Baseline inconsistency and a typo that hints at carelessness. You compare against RTN/RRTN/RVRT in text, but Table 1 spells “RVTN”—not a good look. Also, Table 2 leaves out RVRT for mask analysis “because it doesn’t output masks,” which is convenient but incomplete; evaluate detection quality indirectly (e.g., via reconstructed-error in GT masks) to keep the comparison fair.

**Questions:**

(1) Which optical flow network do you use, how is it trained (on degraded inputs or not), and what is the exact architecture of the guidance-map fusion G (channels, ops, normalization)? Provide ablations with different flow qualities.

(2) Is eq. (8) a typo (L2 instead of L1) or intentional? If intentional, please rename it and re-run the ablation showing the impact of true L1 vs your current term.

(3) Temporal metrics: Report t-LPIPS / warping error / FVD on the synthetic test and a real-film subset. If you claim temporal coherence, please measure it.

---

### Meta-Review · Area_Chair_6NJs · 2025-12-12

**Summary:**

This paper is rejected due to critical flaws in method transparency, evaluation comprehensiveness, and real-world applicability:

- Core components (optical flow model, guidance fusion G(⋅), mask parameters) are underspecified, hindering reproducibility; typos and missing citations further impact presentation quality.
- Evaluation is incomplete: no temporal consistency metrics (e.g., t-LPIPS, FVD) despite coherence claims, over-reliance on synthetic data (limited real-film validation), and lack of comparisons to recent diffusion-based SOTA (e.g., Upscale A Video, SeedVR).
- Methodological limitations (reliance on intact first/last frames, sensitivity to optical flow errors) and unaddressed efficiency (runtime/memory) limit practical use.
- Key hyperparameter ablations (e.g., λ) and perceptual validation (human studies) are absent.

These issues cannot be resolved via minor revisions and fail to meet publication standards. Last but not least, the authors did not provide a rebuttal to address these issues.

**Reviewer Concerns:**

Since the author did not provide a rebuttal to resolve these issues, all the concerns remain.

**Reviewer Scores:**

Since the author did not provide a rebuttal to resolve these issues, all the concerns remain.

---

### Decision · Program_Chairs · 2026-01-26

Reject